# Feasibility, reproducibility and characteristics of coronary bifurcation type assessment by three-dimensional optical coherence tomography

**Takashi Nishimura**[ID]**, Takayuki Okamura\*, Tatsuhiro Fujimura, Yosuke Miyazaki, Hitoshi Takenaka, Hideaki Akase, Hiroki Tateishi, Mamoru Mochizuki, Hitoshi Uchinoumi, Tetsuro Oda, Masafumi Yano**

Division of Cardiology, Department of Medicine and Clinical Science, Yamaguchi University Graduate School of Medicine, Ube, Japan

\* t-okamu@yamaguchi-u.ac.jp

## Abstract

### Aim

To investigate the characteristics of coronary artery bifurcation type (parallel or perpendicular type) using three-dimensional (3D) optical coherence tomography (OCT), and determine the feasibility, reproducibility, assessment time and correlation with bifurcation angles measured by 3D quantitative coronary angiography (QCA).

### Methods and results

We evaluated 60 lesions at the coronary bifurcation that were treated by main vessel (MV) stenting with kissing balloon inflation (KBI) under OCT/optical frequency domain imaging (OFDI) guidance. Inter- and intra-observer agreement regarding the assessment of 3D bifurcation types were 0.88 and 0.94, respectively. The assessment times of 3D-OCT bifurcation type with OCT and OFDI were within about 30 seconds. 3D-OCT bifurcation types showed the greatest correlation with the distal bifurcation angle assessed by 3D-QCA among the three bifurcation angles (distal bifurcation angle, proximal bifurcation angle and main vessel angle), and the optimal cut-off distal bifurcation angle to predict a perpendicular type bifurcation, as determined by ROC analysis, was 51.0˚ (AUC 0.773, sensitivity 0.80, specificity 0.67). Based on this cut-off value for the distal bifurcation angle (51˚), the diagnostic accuracy for perpendicular type bifurcation in cases with a BA $\geq$ 51˚ (n = 34) was 70.6% (24/34) and that of the parallel type bifurcation in cases of BA < 51˚ (n = 26) was 76.9% (20/26).

### Conclusion

Performing 3D-OCT for assessment of coronary artery bifurcation type is feasible and simple, and can be done in a short time with high reproducibility.

**Data Availability Statement:** All relevant data are within the paper and its Supporting information files.

**Funding:** The author received no specific funding for this work.

**Competing interests:** Dr. Okamura received honoraria from Terumo Corp. and Abbott Vascular. The other authors have no conflicts of interest to declare in relation to this investigation.

**Abbreviations:** 2D, two-dimensional; 3D, three-dimensional; BA, bifurcation angle; ISA, incomplete stent apposition; KBI, kissing balloon inflation; MACE, major adverse cardiac event(s); MV, main vessel; OCT, optical coherence tomography; OFDI, optical frequency domain imaging; PCI, percutaneous coronary interventions; QCA, quantitative coronary angiography; SB, side branch.

## Introduction

Coronary bifurcation lesions account for 15%– 20% of percutaneous coronary interventions (PCI) and remain one of the most challenging situations in interventional cardiology in terms of procedural success rates and long-term cardiac events [1]. Bifurcation angle (BA) is one of the factors related to clinical outcomes during and after treatment for bifurcation lesions [2–7]. Although BA can be measured by quantitative coronary angiography (QCA), the accuracy of BA derived from two-dimensional (2D) QCA is limited because of overlapping of branches and vessel foreshortening. Measuring BAs is theoretically more precise with three-dimensional (3D) rather than 2D-QCA [8, 9]. Recently, optical coherence tomography (OCT) has been used as a guide for treating bifurcation lesions [10, 11]. Three-dimensional reconstruction using OCT pullback facilitates understanding of interactions between stents and vessel walls [12]. Bifurcation type, whether perpendicular or parallel, as determined by 3D-OCT, was introduced as a parameter for visually assessing bifurcation appearance and was found to be associated with the incidence of incomplete stent apposition (ISA) at the side branch (SB) ostium after bifurcation PCI [13, 14]. However, the characteristics of bifurcation type on 3D-OCT are not well known. Therefore, we investigated the characteristics of 3D-OCT bifurcation type to determine its feasibility, reproducibility, assessment time and correlation with respect to the 3D-QCA bifurcation angle. In addition, the correlation between 3D-OCT bifurcation type and ISA at the bifurcation segment after single stenting with kissing balloon inflation (KBI) was also evaluated.

## Methods

### Study population

This study included all consecutive patients who underwent PCI for de-novo bifurcation lesions, which had a SB more than 2 mm in diameter by visual assessment, under OCT or Optical Frequency Domain Imaging (OFDI) guidance at Yamaguchi University Hospital between January 2013 and November 2017, using the strategy of a simple single stent crossing over a SB and subsequent KBI. The main exclusion criteria were in-stent restenosis, cardiogenic shock, chronic total occlusion, ST-elevation myocardial infarction and the bifurcation anatomy of trifurcation.

This retrospective study protocol was approved by The Medical Ethics Committee of Yamaguchi University Hospital. All patients had the opportunity to opt-out of the study.

### Quantitative coronary angiography

BAs (distal BA, proximal BA, and main vessel (MV) angle), which were measured in the diastolic phase on 3D images generated from two pre-procedure coronary angiograms that were separated by > 30˚, were measured by QAngio® XA 3D Research Edition, v1.0 software (Medis Specials, Leiden, the Netherlands) [8]. The distal BA was defined as the angle between the distal main branch and side branch, the proximal BA was defined as the angle between the proximal MV and SB, and the MV angle was defined as the angle between the proximal MV and distal main branch [8, 15]. Additionally, the reference lumen diameter, minimum lumen diameter of each branch, and percent diameter stenosis were analyzed using the QAngioXA version 7.3 (Medis Specials, Leiden, the Netherlands) [16].

### Procedure and OCT image acquisition

We acquired OCT pullbacks using an Iluminen™ OCT Imaging system (Abbott Vascular, Santa Clara, CA, USA) with a Dragonfly™ intravascular imaging catheter (Abbott Vascular), or

the Lunawave™ OFDI system (Terumo Corporation, Tokyo, Japan) with a FastView™ intravascular imaging catheter (Terumo Corporation). Automatic pullbacks proceeded at 18 or 36 mm/s for OCT and at 20 or 40 mm/s for OFDI during contrast injection for blood clearance from the lumen area using a power injector. The 3D-OCT/OFDI images acquired from all patients were reconstructed using an online system. Distal and proximal reference site and minimum lumen site were determined from longitudinal and cross-sectional OCT/OFDI images of MV pullback, and distal reference lumen area, proximal reference lumen area and minimum lumen area were analyzed.

### Three-dimensional OCT/OFDI reconstruction and quantitative analysis

3D-OCT bifurcation types were assessed according to whether their bifurcations appeared parallel or perpendicular on pre-procedural 3D-OCT/OFDI images [13]. Briefly, the perpendicular type was defined as a bifurcation in which the SB opening was visible as an elliptical shape and was not concealed by the carina when viewed perpendicular to the vessel wall on the cutaway view of the 3D-OCT/OFDI image. In contrast, parallel type bifurcation was defined as a bifurcation in which the proximal course of the SB was concealed behind the carina [13] (Fig 1). The feasibility, reproducibility, and assessment time of 3D-OCT bifurcation type was also evaluated. To ensure inter-observer reproducibility of the assessment of 3D-OCT bifurcation type, thirty-four randomly selected images were assessed by a dedicated interpreter (T.N.) and an independent second interpreter (T.O.). Thirty-four images were assessed to ensure intra-observer reproducibility of the assessment of 3D-OCT bifurcation type at ≥ 4 weeks after the initial evaluation. Assessment time of the 3D-OCT bifurcation type was defined as the time from the start of 3D reconstruction on the OCT/OFDI console to completing the visual assessment of 3D-OCT bifurcation types. The relationship between 3D-OCT bifurcation type and the BA derived by 3D-QCA was also investigated.

Additionally, post-procedural OCT images were quantified. Frame-by-frame cross-sectional images were analyzed by counting all individual struts on each frame. ISA, including floating struts at the SB ostium, defined as separation of at least one stent strut from the vessel wall, was evaluated. Struts were classified as ISA if the distance between the strut marker and lumen contour exceeded the specific strut thickness plus axial resolution of OCT (14 μm) [17]. Strut apposition was assessed in every frame of the bifurcation segment [18]. All distances were measured in perpendicular cross-sections from OCT pullback in the MV. ISA struts at the bifurcation segment were counted. The incidence of ISA was assessed in terms of either the cut-off value of the distal bifurcation angle in 3D-QCA or the 3D-OCT bifurcation type.

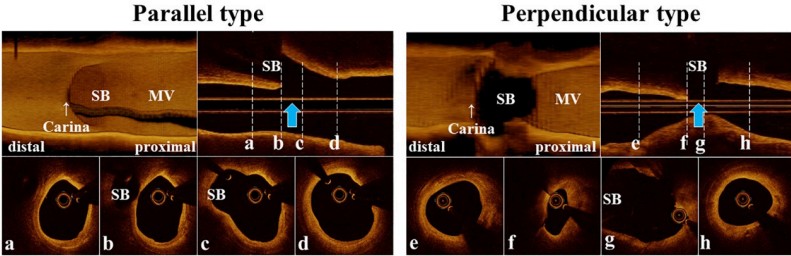

**Fig 1. Definition of 3D-OCT bifurcation type.** Parallel type: The proximal SB course is concealed behind the carina when viewed perpendicular to the vessel wall. Perpendicular type: The visible elliptical SB opening is not concealed by the carina when viewed perpendicular to the vessel wall. Blue arrows, direction of 3D image view. 3D, three-dimensional; SB, side branch, MV, main vessel.

## Statistical analysis

Continuous data are presented as means ± standard deviation (SD) if normally distributed, or as medians and interquartile ranges (IQRs) if not. Continuous variables were compared using Student t-tests if normally distributed and the nonparametric Wilcoxon rank-sum test if not. Categorical variables are presented as numbers and ratios (%) determined in chi-squared tests. Inter- and intra-observer agreement regarding assessments of the 3D-OCT bifurcation type was determined using the kappa statistic. In addition, receiver operating characteristic curve analysis of the distal BA, proximal BA and MV angle was performed to identify the optimal cut-off value of the BAs for prediction of perpendicular type bifurcation. All data were statistically analyzed using JMP software (version 13, JMP Pro, SAS Institute Japan, Tokyo, Japan). All statistical tests were two-sided, and values with $p < 0.05$ were considered significant.

# Results

## Study population

The flow of patient selection in this study is shown in Fig 2. We initially assessed 68 bifurcation lesions in 67 consecutive patients based on the inclusion and exclusion criteria. In the assessment of 3D-OCT bifurcation type, 91.2% (62/68) of the lesions were assessable. In the six cases in which bifurcation type could not be assessed, assessment inability was due to artefacts of the guidewire shadow in four cases and non-uniform rotational distortion (NURD) in two cases. In the measurement of 3D-QCA BA, 97.1% (66/68) of the lesions were assessable. Two cases were not assessable due to difficulties in separating the MV from the SB. Finally, we analyzed 60 bifurcation lesions in 59 patients. Tables 1 and 2 summarize the characteristics of the patients, and characteristics of the lesions and procedures, respectively.

## Assessment time and reproducibility of 3D-OCT bifurcation type

Assessment time of 3D-OCT bifurcation type in OCT and OFDI were 6.9 ± 1.5 seconds and 30.1 ± 4.2 seconds, respectively. Inter- and intra-observer agreement regarding the assessment of 3D-OCT bifurcation types were 0.88 and 0.94 in the kappa statistic, respectively.

## Relationship between 3D-OCT bifurcation types and 3D-QCA BAs

Fig 3 shows the results of ROC curve analysis of the ability of the distal BA, proximal BA and MV angle to predict perpendicular type bifurcation. The optimal cut-off values of distal BA,

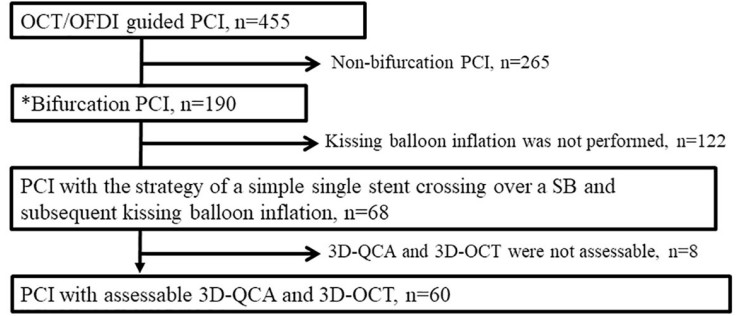

*Bifurcation PCI was defined as cases with a more than 2.0 mm diameter SB or a SB protected by a jailed guidewire.

**Fig 2. Study flow chart.** OCT, optical coherence tomography; OFDI, optical frequency domain imaging; PCI, percutaneous coronary intervention; SB, side branch; 3D, three-dimensional, QCA, quantitative coronary angiography.

**Table 1. Patient characteristics.**

| Variable | n = 60 |
|---|---|
| Age, mean ± SD | 71.2 ± 7.9 |
| Male sex, n (%) | 46 (76.7) |
| Hypertension, n (%) | 46 (76.7) |
| Dyslipidemia, n (%) | 43 (71.7) |
| Diabetes mellitus, n (%) | 27 (45.0) |
| Hemodialysis, n (%) | 4 (6.7) |
| Former smoker, n (%) | 37 (61.7) |
| Clinical presentation | |
| Stable angina pectoris, n (%) | 13 (21.6) |
| Unstable angina pectoris, n (%) | 7 (11.7) |
| Old myocardial infarction, n (%) | 3 (5.0) |
| Silent ischemia*, n (%) | 37 (61.7) |

SD = standard deviation,

*including staged PCI.

proximal BA and MV angle to predict perpendicular type bifurcation on ROC analysis were 51.0˚ (AUC 0.773, sensitivity 0.80, specificity 0.67), 136.0˚ (AUC 0.726, sensitivity 0.50, specificity 0.90), and 137.0˚ (AUC 0.530, sensitivity 0.43, specificity 0.73), respectively. Therefore, the distal BA showed the greatest correlation with 3D-OCT bifurcation types among the three BAs. The cumulative curve for distal BAs is shown in Fig 4. Based on the cut-off value for the distal BA of 51˚, the diagnostic accuracy for perpendicular type bifurcation in cases with a BA of $\geq$ 51˚ (n = 34) was 70.6% (24/34), and that for parallel type bifurcation in cases with a BA of < 51˚ (n = 26) was 76.9% (20/26). Moreover, the concordance rate of 3D-OCT bifurcation types was 61.5% (16/26) in identifying BAs close to the cut-off value (40–59˚), whereas it was 82.4% (28/34) in cases outside this range of distal BA.

With respect to the incidence of ISA at the bifurcation segment, there were no significant differences between distal BAs of < 51˚ and distal BAs of $\geq$ 51˚ (6.1 [2.5–12.2] % vs. 9.1 [3.3–15.9] %, p = 0.30), while there was a significant difference between parallel and perpendicular types (4.3 [1.7–12.2] % vs. 10.0 [4.7–18.2] %, p = 0.014). Representative cases are shown in Fig 5.

## Discussion

The main findings of this study were as follows: 3D-OCT bifurcation type shows the greatest correlation with the distal BA measured by 3D-QCA. Assessment of the 3D-OCT bifurcation type is feasible and can be done visually and easily in a short time with high reproducibility.

### Assessment of BA

BAs can be measured using various methods, including coronary computed tomography angiography (CCTA) and QCA. Previous publications reported that measurements by CCTA reflected real BAs in in vitro experiments [19, 20]. However, CCTA is not performed before the procedure in all cases. 3D-QCA requires dedicated bifurcation software. OCT can be used as a guide for stent optimization [21, 22]. Three-dimensional OCT/OFDI during PCI allowed simple and rapid visual assessment of the appearance of the bifurcation immediately after 3D reconstruction of bifurcation lesions, with high reproducibility. Although the assessment time

**Table 2. Lesion and procedure characteristics.**

| Variable | n = 60 |
|---|---|
| Treated bifurcation, n (%) | |
| LM | 34 (56.7) |
| LAD | 16 (26.7) |
| LCx | 6 (10.0) |
| RCA | 4 (6.7) |
| Medina classification, n (%) | |
| (1,1,1) | 1 (1.7) |
| (1,1,0) | 22 (36.7) |
| (1,0,0) | 1 (1.7) |
| (0,1,1) | 3 (5.0) |
| (0,1,0) | 33 (55.0) |
| Stent type, n (%) | |
| BMX-J | 24 (40.0) |
| Resolute integrity | 16 (26.7) |
| Ultimaster | 9 (15.0) |
| Synergy (Promus Premier) | 11 (18.3) |
| Stent size, mm | 3.5 [2.5–3.5] |
| Stent length, mm | 22.0 [18.0–25.5] |
| Kissing balloon inflation | |
| MV balloon size, mm | 3.0 [2.75–3.5] |
| SB balloon size, mm | 2.5 [2.0–3.0] |
| Rotational atherectomy, n (%) | 6 (10.0) |
| Pre-dilatation, n (%) | 57 (95.0) |
| Post-dilatation, n (%) | 59 (98.3) |
| POT, n (%) | 34 (56.7) |
| Distal recrossing, n (%) | 56 (93.3) |
| Pre-procedural QCA | |
| Proximal main vessel | |
| Ref. lumen diameter, mm | 3.1 [2.55–3.7] |
| Min. lumen diameter, mm | 2.5 [2.0–3.1] |
| % Diameter stenosis | 15.1 [5.9–35.2] |
| Distal main vessel | |
| Ref. lumen diameter, mm | 2.0 [1.7–2.6] |
| Min. lumen diameter, mm | 1.3 [0.8–1.9] |
| % Diameter stenosis | 30.0 [21.0–56.4] |
| Side branch | |
| Ref. lumen diameter, mm | 2.4 [1.9–3.1] |
| Min. lumen diameter, mm | 2.1 [1.6–2.7] |
| % Diameter stenosis | 13.7 [8.4–21.5] |
| Pre-procedural OCT analysis | |
| Proximal ref. lumen area, mm$^2$ | 9.3 [6.6–12.7] |
| Distal ref. lumen area, mm$^2$ | 5.3 [3.6–6.9] |
| Minimum lumen area, mm$^2$ | 1.8 [1.2–2.6] |
| OCT/OFDI system, pullback speed | |
| OCT, 18 mm/s | 9 (15.0) |
| 36 mm/s | 25 (41.7) |
| OFDI, 20 mm/s | 10 (16.7) |

(*Continued*)

**Table 2.** (Continued)

| Variable | n = 60 |
|---|---|
| 40 mm/s | 16 (26.7) |
| Flushing material | |
| Contrast | 60 (100.0) |

LM, left main; LAD, left anterior descending; LCx, left circumflex; RCA, right coronary artery; MV, main vessel; SB, side branch; POT, proximal optimization technique; QCA, quantitative coronary angiography; Ref, reference; Min, minimum; OCT, optical coherence tomography; OFDI, optical frequency domain imaging.

in OFDI was longer than that when using the OCT system in the present study, it was acceptable during PCI [23].

## Clinical significances of BA and 3D-OCT bifurcation type

Clinical significances of the coronary BA have been discussed in previous numerical reports. Because of the association between the 3D-OCT bifurcation type and the distal BA, the information of 3D-OCT bifurcation type (parallel or perpendicular) before stenting might help us to determine the treatment strategy in bifurcation lesions.

Watanabe et al. found a strong correlation between the carina-tip angle, which was defined as the angle between the MV lumen contour line and the SB lumen contour line at the carinal surface in OCT images, and the BA derived from CCTA [24]. Moreover, a narrower carina-tip angle (less than 50˚) was an independent predictor of SB complications, defined as angiographic worsening of SB stenosis (>75%) after OCT-guided bifurcation stenting [25]. It is curious that our cut-off BA of 51˚ for discriminating between parallel and perpendicular type bifurcations was very similar to the narrower carina-tip angle for predicting SB compromise in this previous study.

Chen et al. reported that the 1-year MACE rate in the DK crush group was significantly lower than that in the Culotte group among patients with distal BA $\geq$ 70˚ [26], which was classified as the perpendicular type bifurcation. Moreover, it has been reported that a significant negative correlation was found between the uncovered strut percentage at the bifurcation segment and distal BA (r = -0.41, P = 0.0024) [24]. With respect to the ISA including the jailing struts at the bifurcation segment after KBI, the distal rewiring into the jailed SB could reduce ISA in the parallel type bifurcation whereas ISA was not always reduced by distal rewiring in the perpendicular type [14]. Besides ISA at the bifurcation segment, 3D-OCT bifurcation type might also be related to SB compromise and restenosis after PCI. Further investigations are warranted to confirm this.

## Differences in the assessment of BA between 3D-QCA and 3D-OCT

The present study showed a discrepancy in the incidence of ISA at the bifurcation segment when using the cut-off value of the distal BA in 3D-QCA evaluation versus the 3D-OCT bifurcation type. We speculate that this phenomenon might be caused by vessel straightening due to the guidewire and by OCT/OFDI catheter insertion into the MV. In other words, 3D-OCT bifurcation type might reflect the relationship between the MV and SB takeoff after stent deployment, almost like predicting the appearance of vessel straightening after stent deployment in advance, because 3D-OCT images were obtained while keeping the vessel straight. In contrast, we hypothesized that 3D-QCA reflects the natural coronary BA. Therefore, 3D-OCT bifurcation type rather than BA on 3D-QCA might better reflect the BA after stent deployment

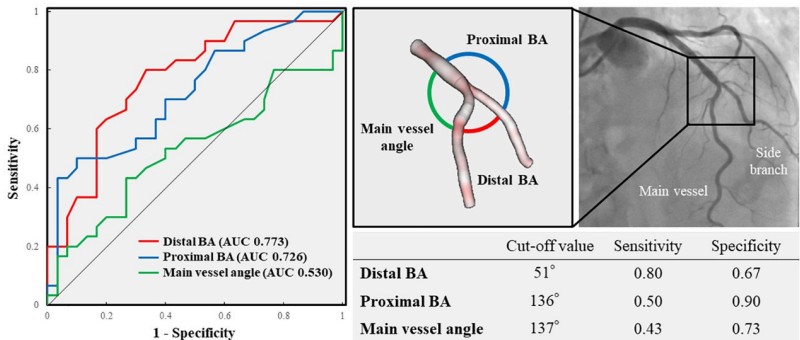

**Fig 3. Receiver operating characteristic curve analysis to predict perpendicular type coronary bifurcation.** AUC, area under the curve; BA, bifurcation angle.

into the MV, and might be associated with ISA at the bifurcation segment after final KBI. Further studies are warranted to confirm these theories.

## Limitations

This study has some limitations. First, this study only included patients who were treated by single stenting with KBI, and bifurcation lesions that were left untreated or were treated by other techniques were not evaluated in this study. This might have influenced the cut-off angle used to predict perpendicular type bifurcations. However, in a clinical setting, assessment of the BA is needed for PCI. Second, we compared 3D-OCT with 3D-QCA, which is more accurate than 2D-QCA. However, BA measured by 3D-QCA might be different from the actual BA. Moreover, this study included both OCT and OFDI. We did not compare these two different assessment systems. Use of IVUS and OCT is recommended for stent optimization during the bifurcation stenting in the current guideline [27]. IVUS also might predict the side branch complication and the bifurcation angle. The previous report showed that the "eyebrow" sign associated with narrower angiographic angles was a powerful predictor of the SB damage after stent implantation in the MV [28]. Further study to investigate the superiority between both modalities may be warranted. Finally, we did not investigate other clinical implications of 3D-OCT bifurcation type (i.e., prediction of SB compromise, clinical outcomes, etc.) except

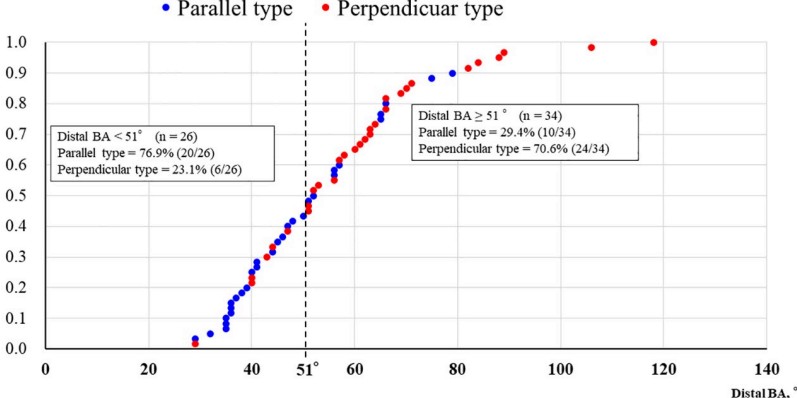

**Fig 4. Cumulative curve analysis for the distal bifurcation angle.** The vertical dotted line indicates the cut-off angle of 51˚ for prediction of perpendicular type bifurcation. BA, bifurcation angle.

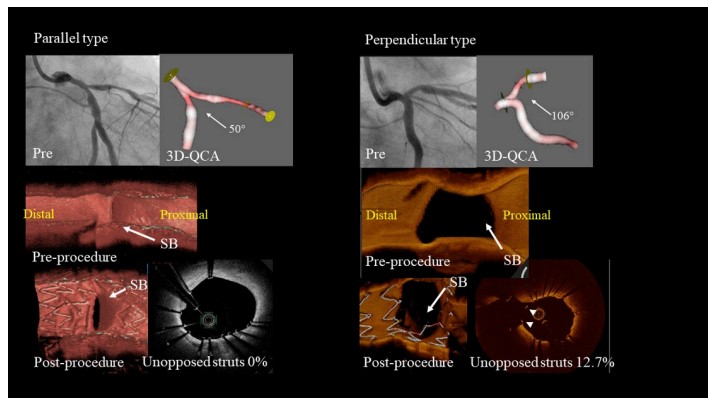

**Fig 5. Representative 3D-OCT bifurcation types.** Parallel type: Jailed stent struts at the SB ostium can be attached towards the SB wall by KBI in parallel type bifurcations in which the distal bifurcation angle is 50˚. Perpendicular type: Jailed stent struts at the SB ostium cannot be sufficiently dilated to fit the elliptical SB ostial rim in perpendicular type bifurcations with a distal bifurcation angle of 106˚. OCT, optical coherence tomography, SB, side branch, KBI, kissing balloon inflation, QCA, quantitative coronary angiography.

for ISA at the bifurcation segment after KBI, and we did not compare the ISA at bifurcation between the parallel type and the perpendicular type due to difficulties adjusting many confounders (S1 Table). Further studies are needed to address this.

## Conclusions

Visual assessment of 3D-OCT bifurcation type is feasible and can be easily performed in a short time with high reproducibility. Moreover, it correlates well with the BA measured by 3D-QCA and might be useful to assess BA during PCI.

## Supporting information

**S1 Table. Supplementary table.** SD = standard deviation, *including staged PCI, LM, left main; LAD, left anterior descending; LCx, left circumflex; RCA, right coronary artery; MV, main vessel; SB, side branch; POT, proximal optimization technique; QCA, quantitative coronary angiography; Ref, reference; Min, minimum; OCT, optical coherence tomography.
(DOCX)

**S1 Data. Analysis data.**
(XLSX)

## Author Contributions

**Conceptualization:** Takashi Nishimura, Takayuki Okamura.

**Data curation:** Takashi Nishimura, Takayuki Okamura, Tatsuhiro Fujimura, Yosuke Miyazaki, Hitoshi Takenaka, Hideaki Akase, Hiroki Tateishi, Mamoru Mochizuki, Hitoshi Uchinoumi, Tetsuro Oda.

**Formal analysis:** Takashi Nishimura, Hitoshi Takenaka, Hideaki Akase.

**Funding acquisition:** Takayuki Okamura.

**Investigation:** Tatsuhiro Fujimura, Yosuke Miyazaki, Hitoshi Takenaka, Hideaki Akase, Hiroki Tateishi, Mamoru Mochizuki, Hitoshi Uchinoumi, Tetsuro Oda, Masafumi Yano.

**Project administration:** Takayuki Okamura.

**Supervision:** Masafumi Yano.

**Writing – original draft:** Takashi Nishimura.

**Writing – review & editing:** Takayuki Okamura, Tatsuhiro Fujimura, Yosuke Miyazaki, Masafumi Yano.

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
