## [Decision Letter · Decision Letter 0]

4 Oct 2021

PONE-D-21-25048

Feasibility, reproducibility and characteristics of coronary bifurcation type assessment by three-dimensional optical coherence tomography

PLOS ONE

Dear Dr. Okamura,

Thank you for submitting your manuscript to PLOS ONE. After careful consideration, we feel that it has merit but does not fully meet PLOS ONE’s publication criteria as it currently stands. Therefore, we invite you to submit a revised version of the manuscript that addresses the points raised during the review process.

Please respond the problems pointed out by reviewers.

We look forward to receiving your revised manuscript.

Kind regards,

Yoshiaki Taniyama, MD, PhD

Academic Editor

PLOS ONE

Journal Requirements:

2. Please provide additional details regarding participant consent. In the Methods section, please ensure that you have specified (1) whether consent was informed and (2) what type you obtained (for instance, written or verbal). If your study included minors, state whether you obtained consent from parents or guardians. If the need for consent was waived by the ethics committee, please include this information.

3. Please be informed that Dr. Okamura received honoraria from Terumo corp and Abbott vascular. The other authors have no conflict of interest to declare with the manuscript from the cover letter. This information should be added to the COI.

Reviewers' comments:

Reviewer's Responses to Questions

**Comments to the Author**

1. Is the manuscript technically sound, and do the data support the conclusions?

Reviewer #1: Partly

Reviewer #2: Yes

2. Has the statistical analysis been performed appropriately and rigorously? 

Reviewer #1: Yes

Reviewer #2: Yes

3. Have the authors made all data underlying the findings in their manuscript fully available?

Reviewer #1: Yes

Reviewer #2: Yes

4. Is the manuscript presented in an intelligible fashion and written in standard English?

Reviewer #1: Yes

Reviewer #2: Yes

5. Review Comments to the Author

Reviewer #1: The authors reported the feasibility and reproducibility of online visual 3D-OCT assessment of bifurcation lesion . 3D-OCT bifurcation type is tentatively correlated with the distal bifurcation angle measured by 3D-QCA. This manuscript is well written and organized. However, following issues need to be addressed.

1. It is unclear what will be the clinical implication of the bifurcation type. Debate persists regarding the optimal treatment of side branch bifurcation lesions including assessment of clinical significance. How can the information of bifurcation type before stenting help us to determine the treatment strategy in bifurcation lesion? Basically, it is not a valuable publication unless the authors demonstrate the clinical implication of this study. Please discuss about that.

2. The authors showed a correlation with the distal bifurcation angle measured by 3D-QCA. However, the clinical impact on bifurcation angle before stenting is limited. I believe that it is more important to acquire optimal projection detected by coronary CT or functional assessment using FFR/FFRCT/QFR than the recognition of bifurcation angle or type before procedure.

3. The authors did not describe the difference in lesion characteristic between parallel type and perpendicular. Is there any differences in lesion characteristics such as distal recrossing or appropriate POT or KBI which is the predictor of side branch complication? Please clarify this.

Reviewer #2: The manuscript entitled “Feasibility, reproducibility and characteristics of coronary bifurcation type assessment by three-dimensional optical coherence tomography” investigates the possible role of 3D-OCT to assess bifurcation angles as compared to 3D-QCA.

The manuscript is overall well written and the topic is interesting. Please find below my comments to improve the manuscript quality:

1) According to authors data, 3D OCT seems to be reliable for the detection of parallel and perpendicular bifurcation angles (BA). However, a binary classification of bifurcation angles into two types only might be poorly accurate and 3D OCT might be not accurate in identifying less overt angles (e.g 40-60°). It would be advisable to discuss this point or to include it in the limitations paragraph.

2) Authors justify the use of 3D OCT for the BA as 3d OCT represents also a useful tool for stent optimization. However, in several centres also IVUS is used for this purpose. Please discuss the possible role of IVUS in such context.

3) As reported by the authors, 51° is the cut-off for side branch complications. It would be interesting to add any possible complications in the parallel and perpendicular BAs, as well as in those with angles > and <51°.

4) Although the readership might be familiar with the topic, it would useful to add a figure explaining the three different BAs (distal, proximal and main vessel).

6. PLOS authors have the option to publish the peer review history of their article (what does this mean?). If published, this will include your full peer review and any attached files.

Reviewer #1: No

Reviewer #2: **Yes: **Giulio Russo

---

## [Author Response · Author response to Decision Letter 0]

17 Nov 2021

We thank the reviewers for carefully reading for our manuscript and for the reviewers’ comments, which have helped us to further improve the quality of our manuscript. We responded to the reviewers’ comments in ‘’Response to Reviewers’’ file. Please confirm the attached document.

---

## [Decision Letter · Decision Letter 1]

17 Jan 2022

Feasibility, reproducibility and characteristics of coronary bifurcation type assessment by three-dimensional optical coherence tomography

PONE-D-21-25048R1

Dear Dr. Okamura,

We’re pleased to inform you that your manuscript has been judged scientifically suitable for publication and will be formally accepted for publication once it meets all outstanding technical requirements.

Kind regards,

Yoshiaki Taniyama, MD, PhD

Academic Editor

PLOS ONE

Additional Editor Comments (optional):

The author responded the problems.

Reviewers' comments:

Reviewer's Responses to Questions

**Comments to the Author**

1. If the authors have adequately addressed your comments raised in a previous round of review and you feel that this manuscript is now acceptable for publication, you may indicate that here to bypass the “Comments to the Author” section, enter your conflict of interest statement in the “Confidential to Editor” section, and submit your "Accept" recommendation.

Reviewer #2: All comments have been addressed

2. Is the manuscript technically sound, and do the data support the conclusions?

Reviewer #2: Yes

3. Has the statistical analysis been performed appropriately and rigorously? 

Reviewer #2: Yes

4. Have the authors made all data underlying the findings in their manuscript fully available?

Reviewer #2: Yes

5. Is the manuscript presented in an intelligible fashion and written in standard English?

Reviewer #2: Yes

6. Review Comments to the Author

Reviewer #2: All issues raised during the revision process have been properly addressed and the manuscript is now suitable for publication

7. PLOS authors have the option to publish the peer review history of their article (what does this mean?). If published, this will include your full peer review and any attached files.

Reviewer #2: **Yes: **Giulio Russo

---

## [Editor Report · Acceptance letter]

21 Jan 2022

PONE-D-21-25048R1 

Feasibility, reproducibility and characteristics of coronary bifurcation type assessment by three-dimensional optical coherence tomography 

Dear Dr. Okamura:

I'm pleased to inform you that your manuscript has been deemed suitable for publication in PLOS ONE. Congratulations! Your manuscript is now with our production department. 

Kind regards, 

on behalf of

Dr. Yoshiaki Taniyama 

Academic Editor

PLOS ONE